

# Effect of next-nearest neighbor hopping
# on the single-particle excitations at finite temperature

**Harun Al Rashid and Dheeraj Kumar Singh**⋆

Department of Physics and Materials Science,
Thapar Institute of Engineering and Technology,
Patiala-147004, Punjab, India

⋆ dheeraj.kumar@thapar.edu

## Abstract

In the half-filled one-orbital Hubbard model on a square lattice, we study the effect of next-nearest neighbor hopping on the single-particle spectral function at finite temperature using an exact-diagonalization + Monte-Carlo based approach to the simulation process. We find that the pseudogap-like dip, existing in the density of states in between the Néel temperature $T_N$ and a relatively higher temperature scale $T^*$, is accompanied with a significant asymmetry in the hole- and particle-excitation energy along the high-symmetry directions as well as along the normal-state Fermi surface. On moving from $(\pi/2, \pi/2)$ toward $(\pi, 0)$ along the normal state Fermi surface, the hole-excitation energy increases, a behavior remarkably similar to what is observed in the $d$-wave state and pseudogap phase of high-$T_c$ cuprates, whereas the particle-excitation energy decreases. The quasiparticle peak height is the largest near $(\pi/2, \pi/2)$ whereas it is the smallest near $(\pi, 0)$. These spectral features survive beyond $T_N$. The temperature window $T_N \lesssim T \lesssim T^*$ shrinks with an increase in the next-nearest neighbor hopping, which indicates that the next-nearest neighbor hopping may not be supportive to the pseudogap-like features.

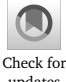

# 1 Introduction

Cuprates are archetypal systems of materials, which besides showing uncoventional superconductivity, also exhibit a wide variety of interesting but complex phases as a function of doping and temperature [1, 2]. Cu atom in the undoped/parent cuprates has $3d^9$ outermost electronic configuration, which gets altered to $3d^9\underline{L}$ as a hole is doped [3]. $\underline{L}$ indicates the fact that the doped hole resides instead on the neighboring oxygen atoms, which forms a bridge between the two neighboring $Cu^{2+}$ as well as a square that surrounds a $Cu^{2+}$ ion. The hole binds to the $Cu^{2+}$ ion leading to the emergence of Zhang-Rice singlet [4,5]. The electron doping, whereas, modifies the electronic configuration from $3d^9$ to $3d^{10}$. The low-energy physics involving either hole or particle doping have been studied quite extensively in the span of last three decades within the one-orbital Hubbard model, which has provided us with important insight into the understanding of correlation effects [6–11].

Photoemission studies at a temperature well above the Néel temperature $T_N$ in the undoped cuprates such as $Sr_2CuO_2Cl_2$ [12] shows characteristics of the quasiparticle excitations, which have several features similar to the one observed in the pseudogap and $d$-wave superconducting phases. First, the quasiparticle peak is sharp near $(\pi/2, \pi/2)$ and gets broadened on approaching $(\pi, 0)$. Secondly, the hole excitation energy is the least near $(\pi/2, \pi/2)$ while it is the largest close to $(\pi, 0)$. The broad peak near $(\pi, 0)$ gets sharpened on doping hole. One-orbital Hubbard model with only nearest neighbor hopping or $t$-$J$ model couldn't reproduce the features in different studies based on different approaches. A crucial role of long-range hopping was emphasized on later, especially, the next-nearest neighbor hopping within $t$-$J$ model often employed to study the hole dynamics [13–16].

A striking feature of the doping-vs-temperature phase diagram of the high-$T_c$ cuprates is the asymmetry with respect to hole or electron doping [1, 17]. In particular, the long-range antiferromagnetic order (AFM) is found to survive up to only a small hole doping of $\sim 1\%$ [18] whereas it is robust against a relatively larger electron doping of $\sim 15\%$ [19]. On the other hand, the $d$-wave superconductivity as well as the pseudogap phase exist in a comparatively wide range of hole doping. Recent experiments suggest that the pseudogap phase found upon hole doping may be marked with the presence of variety of symmetry breaking phonomena including the nematic order [20], stripe order [21–23], short- or long-range charge-density wave [24–27], pair-density wave [28] etc., while the possibility of coexisting more than one of these is also not ruled out.

The one-orbital Hubbard model with only nearest-neighbor hopping $t$ possesses particle-hole symmetry, therefore it cannot describe the asymmetrical behavior of the phase diagram. Earlier works suggest a crucial role for the next-nearest neighbor hopping parameter $t'$ [13, 14], which allows the hole/electron to hop within the same sublattice, in describing various spectral features [12], spin-wave excitation spectra [29,30], and the asymmetry of the phase diagram. For the hole and electron doping, $t'$ is positive and negative, respectively, thus bringing in frustration in the case of former. In presence of next-nearest neighbor hopping, the AFM state is known to be stabilized for a wide electron-doping region, whereas even a single hole doping may prove to be detrimental to it [31]. Besides, the asymmetry introduced in the density of states, $t'$ is also known to enhance the tendency towards ferromagnetic order (FM) upon hole or electron doping [32]. Furthermore, the maximum value of the transition temperature $T_c$ for the high-$T_c$ cuprates may exhibit sensitiveness to $t'$ [33].

The spectral properties in the half-filled Hubbard model has been investigated by a variety of methods. While the slave-boson [34] or -spin [35] meanfield theoretic approach captures the Mott transition, they fail to incoporate the spatial fluctuations in the order-parameter fields. The methods, which go beyond mean-field theories, such as dynamial-mean field theories (DMFT) [36,37], cluster-perturbation theory (CPT) [38,39], determinant quantum Monte

Carlo (QMC) [7,40–42] etc. suffer from finite-cluste size induced momentum resolution. Furthermore, the QMC-based simulations face the sign problem in the absence of particle-hole symmetry when the next-nearest neighbor hopping is taken into account. Even for the doping away from half-filling, these methods may be applicable for only a certain temperature range. For these reasons, they have been employed mostly for the Hubbard model at half filling with only nearest neighbor hopping. The spectral features, in the hole-doped cases, were examined using the Hartree-Fock meanfield [43], DMFT [44, 45], QMC [11, 46, 47], classical Monte Carlo [48, 49], Gutzwiller approximation [50] etc., where some of them focused on the correlation corresponding to the $d$-wave superconductivity as well. Most of the work using variational Monte Carlo (VMC) has been largely restricted for obtaining the ground-state phase diagram [51–53], however, a recent attempt explores the spectral features but only without next-nearest neighbor hopping [11]. In the absence of second-neighbor hopping, the nature of hole-excitation energy may retain its features even beyond $T_N$ with the peak height almost independent of the momentum along the normal-state Fermi surface [54]. The peak-to-peak distance increases on going from $(\pi/2, \pi/2)$ to $(\pi, 0)$ along the normal state Fermi surface.

Not much is known about the variation of the single-particle spectral features with temperature when $t'$ is incorporated into a microscopic model such as Hubbard model at half filling. Does the peak-to-peak separation increase on including $t'$? If the hole-excitation energy increases on moving from $(\pi/2, \pi/2)$ to $(\pi, 0)$ then how does the particle-excitation energy vary? How do these spectral features evolve with change in temperature? Answers to these questions are of significant interest in order to understand the role of $t'$ on the pseudogap-like behavior.

In this paper, we investigate the role of nearest-neighbor hopping on the single-particle spectral function within one-orbital and half-filled Hubbard model as a function of temperature. We employ exact-diagonalization + Monte-Carlo scheme based on parallelization to extract the characteristics of single-particle spectral function at different temperature. In order to handle a larger system size so that the momentum resolution without any finite-size effect can be achieved, traveling-cluster approximation (TCA) [55] and twisted-boundary condition [56] are used additionally. We arrive at the following major results for the single-particle spectral function: on moving along the normal state Fermi surface from $(\pi/2, \pi/2)$ to $(\pi, 0)$, (i) the hole- and particle-excitation energy increases and decreases, respectively and (ii) the height of the quasiparticle peak for the hole- and particle excitation decreases and increases, respectively. (iii) Below $T_N$, the spectral weight is significantly suppressed along $(0, 0) \rightarrow (\pi/2, \pi/2)$ and $(\pi, 0) \rightarrow (0, 0)$ for the upper band and along $(\pi/2, \pi/2) \rightarrow (\pi, \pi)$ and $(\pi, \pi) \rightarrow (\pi, 0)$ for the lower band. (iv) For $T \gtrsim T_N$, a relatively larger spectral weight near $(\pi/2, \pi/2)$ is continued to be noticed in comparison to $(\pi, 0)$. (v) The hole-excitation energy increases with $t'$ and (vi) the dip in the density of states, which persists beyond $T_N$ becomes shallower with increasing $t'$ indicating that the latter may be unfavorable for the pseudogap-like features, which is also reflected in the behavior of momentum-resolved spectral function.

## 2 Model and method

We start with the following one-orbital Hubbard Hamiltonian

$$\mathcal{H} = \sum_{\mathbf{i}, \delta, \sigma} t_{\mathbf{i}, \mathbf{i}+\delta} d_{\mathbf{i}\sigma}^{\dagger} d_{\mathbf{i}+\delta\sigma} - \mu \sum_{\mathbf{i}, \sigma} n_{\mathbf{i}\sigma} + U \sum_{\mathbf{i}} n_{\mathbf{i}\uparrow} n_{\mathbf{i}\downarrow}, \tag{1}$$

where the operator $d_{i\sigma}^{\dagger}$ ($d_{i\sigma}$) creates (annihilates) an electron with spin $\sigma$. $\delta$ is a vector which connects a given site to the nearest neighboring and next-neighboring sites. $t_{i,i+\delta} = -t$ and $t'$ for the nearest and next-nearest neighbor hopping, respectively. $n_{i\sigma} = d_{i\sigma}^{\dagger} d_{i\sigma}$ is the charge-

density operator for spin $\sigma$ electron. $U$ and $\mu$ are the onsite Coulomb repulsion and chemical potential, respectively.

For the simulation, the grand-partition function used corresponding to the original Hamiltonian given by Eq. (1) is

$$\mathcal{Z} = \int \mathcal{D}\psi \mathcal{D}\bar{\psi} e^{-\mathcal{A}[\psi,\bar{\psi}]}, \tag{2}$$

where the action [57]

$$\mathcal{A} = \int_0^\beta d\tau \sum_{\mathbf{i},\delta,\sigma} \bar{\psi}_{i\sigma}(\tau)((\partial_\tau - \mu)\delta_{\mathbf{i},\mathbf{i}+\delta} + t_{\mathbf{i},\mathbf{i}+\delta})\psi_{\mathbf{i}+\delta\sigma}(\tau) + U\sum_{\mathbf{i}}\left(\frac{n_{\mathbf{i}}^2(\tau)}{4} - \left(\vec{S}_{\mathbf{i}}(\tau)\cdot\hat{\mathbf{r}}_i\right)^2\right). \tag{3}$$

$\psi_{i\sigma}(\tau)$ and $\bar{\psi}_{i\sigma}(\tau)$ are the Grassman variables corresponding to the operators $d_{i\sigma}$ and $d_{i\sigma}^\dagger$, respectively. The form of interaction term used in Eq. (3) follows from

$$n_{i\uparrow}n_{i\downarrow} = \frac{n_i^2}{4} - (S_{iz})^2. \tag{4}$$

$S_{i\mu} = \frac{1}{2}\sum_{\alpha,\beta} d_{i\alpha}^\dagger \sigma_{\alpha\beta}^\mu d_{i\beta}$ is the $\mu^{th}$ component of local electron-spin operator. $S_{iz} = \vec{S}_i \cdot \hat{\mathbf{r}}$ when the unit vector $\hat{\mathbf{r}}$ is oriented along $z$ axis. The Hubbard interaction has the $SU(2)$ rotational symmetry in spin space, therefore

$$n_{i\uparrow}n_{i\downarrow} = \frac{n_i^2}{4} - (\vec{S}_i \cdot \hat{\mathbf{r}}_i)^2, \tag{5}$$

where the unit vector $\hat{\mathbf{r}}$ may now be oriented along any arbitrary direction.

To make further progress, we use Hubbard-Stratonovich (HS) transformation to decouple the Hubbard interaction by introducing two auxiliary fields, a scalar field $\phi_i(\tau)$ coupled to the charge density $n_i$ and a vector field $\mathbf{m}_i$ coupled to the spin of electron $\sigma_i$. This modifies the grand-partition function to

$$Z = \int \prod_i \frac{d\bar{\psi}_i d\psi_i d\phi_i d\mathbf{m}_i}{4\pi^2 U} e^{-\mathcal{A}(\bar{\psi}_i,\psi_i,\phi_i,\mathbf{m}_i)}, \tag{6}$$

with contribution to the action due to the on-site interaction being

$$\mathcal{A}_{int} = \int_0^\beta d\tau \sum_{\mathbf{i}} \left\{ i\phi_{\mathbf{i}}\sum_\sigma \bar{\psi}_{\mathbf{i}\sigma}\psi_{\mathbf{i}\sigma} - \mathbf{m}_{\mathbf{i}}\cdot\sum_{\sigma\sigma'} \bar{\psi}_{\mathbf{i}\sigma}\vec{\sigma}_{\sigma\sigma'}\psi_{\mathbf{i}\sigma'} \right\} + \frac{1}{U}\sum_{\mathbf{i}}\left(\phi_{\mathbf{i}}^2 + \mathbf{m}_{\mathbf{i}}^2\right). \tag{7}$$

In the simulation, the Hubbard-Stratonovich fields are treated as classical fields so that time ($\tau$) dependence is ignored. Next, we use the saddle-point approximation for the scalar field $\phi_i$, for which, the spatial and temporal fluctuations are ignored, which yields $(U/2)\langle n_i \rangle = U/2$ at half filling. Thus, the effective Hamiltonian can be shown to be

$$H_{eff} = \sum_{\mathbf{i},\delta,\sigma} t_{\mathbf{i},\mathbf{i}+\delta} d_{\mathbf{i}\sigma}^\dagger d_{\mathbf{i}+\delta\sigma} - \tilde{\mu}\sum_i n_i - \frac{U}{2}\sum_i \mathbf{m}_i \cdot \vec{\sigma}_i + \sum_i \frac{U}{4}\mathbf{m}_i^2$$
$$= H_e + H_{cls}, \tag{8}$$

where $\tilde{\mu} = \mu - U/2$. The field $\mathbf{m}_i$ are scaled by $\mathbf{m}_i \to \frac{U}{2}\mathbf{m}_i$ so that it can be turned into a dimensionless field. Note the term in the effective Hamiltonian $H_{cls} = \frac{U}{4}\mathbf{m}_i^2$, which is treated as classical in the simulation process, i.e., the temporal fluctuations are ignored. On the other hand, all the spatial and thermal fluctuations are retained as described below.

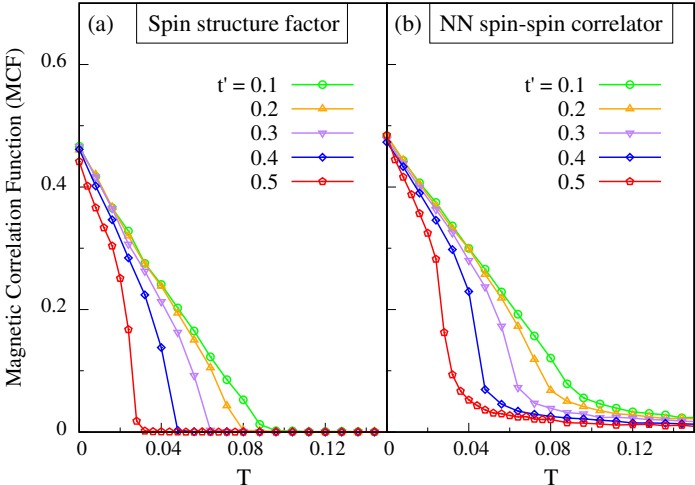

Figure 1: The (a) long- and (b) short-range AFM correlations as a function temperature for different $t' = 0.1, 0.2, 0.3, 0.4$, and $0.5$. The long-range correlation function shows relatively sharper rise for larger $t'$. The short-range correlation function as defined in the text does not vanish beyond $T_N$ and it may show weak dependence on $t'$.

The equilibrium configurations for the auxiliary field $\{\mathbf{m}_i\}$ are generated according to the following distribution

$$P\{\mathbf{m}_i\} \propto Tr_{d,d^\dagger} e^{-\beta H_e} e^{-\beta H_{cls}}, \tag{9}$$

where the trace over the fermionic degree of freedom cannot be calculated exactly due to the terms in $H_e$ coupled to the classical fields. Therefore, the equilibrium field is generated through MC sampling. In each MC update process, $H_e$ is diagonalized and then the eigenvalues are used to calculate the change in free energy of the system. ED + MC method allows an access only to a small system size. The observables generated in the simulation suffer from the finite-size effect and therefore limiting the access to a good momentum resolution.

The limitation posed for the momentum resolution can be overcome upon combining three steps in the simulation process. For each update process, instead of considering the full lattice, only a small cluster of sites around the update site is considered. Thus, the process involves the diagonalization of Hamiltonian for the cluster (size $N_c \times N_c$) centered around the update site. The computational cost is reduced by a factor of $\sim N_c/N_l$, where $N_l \times N_l$ is the original system size [55]. We use $N_l = 40$ and $N_c = 8$ throughout the current work. The simulation can be further sped up by using parallelized update process, where $N_p$, a factor of total number of available processors, sites can be updated simultaneously. The computational cost reduction in this step is achieved up to a factor of $\sim 1/N_p$ [54, 58]. In order to reduce the finite size effect further, we make use of twisted-boundary condition (TBC) [56], where a superlattice is formed by repeating the original system of size $N_l \times N_l$ and associated field in $x$- and $y$-direction $N_t$ times. The spectral function calculated for such a superlattice is equivalent to the spectral function of an effective system size $N_l N_t \times N_l N_t$. In order to calculate the spectral function, which is to be discussed later, we use $N_t = 6$ which allows us to access an effective system size of $240 \times 240$. It may be noted that, in the absence of particle-hole symmetry ($t' \neq 0$), the chemical potential fluctuates with temperature, therefore, the simulation requires to check the chemical potential and update the same in accordance with half filling at intermediate steps. This enhances the computational cost of the simulation. Another factor which is responsible for raising of the computational cost is the frustration introduced by the next-nearest neighbor hopping in the system, which delays the thermal equilibration.

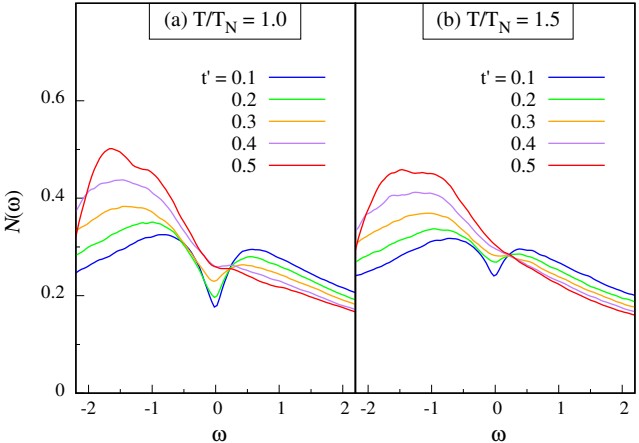

Figure 2: The DOS as a function of energy $\omega$ at different temperatures $T/T_N = $ (a) 1.0 and (b) 1.5 for various $t' = 0.1, 0.2, 0.3, 0.4$ and 0.5. Unlike $t' = 0$ case, the dip in the DOS may vanish completely at higher temperature when the next-nearest neighbor hopping is incorporated.

The simulation process is started at a temperature, which is nearly twice of $T_N$, and then the system is cooled down in small steps of temperature. At each temperature, first thousand MC sweeps are used to reach equilibrium field configuration $\{\mathbf{m}_i\}$. In the next thousand sweeps, data related to structure factor, spectral function etc. are obtained for different thermal configurations so as to carry out thermal averaging. We set $U$ to be $4t$, which is not far from the screened value as recent works suggest [59]. Since the Hubbard model can be mapped to the Heisenberg model with the exchange coupling $4t^2/U$, it can be noted that with larger $U$, one expects a larger broadening in the spectral function. This follows from the softening of the AFM state with increasing $U$.

# 3 Results

Fig. 1(a) shows the structure factor for the AFM state with ordering wavevector $\mathbf{Q} = (\pi, \pi)$ given by

$$S(\mathbf{Q}) = \frac{1}{N^2} \sum_{\mathbf{i}, \mathbf{j}} \langle \mathbf{m_i} \cdot \mathbf{m_j} \rangle e^{i\mathbf{Q} \cdot (\mathbf{r_i} - \mathbf{r_j})}, \tag{10}$$

where $\mathbf{r_i}$ is the position vector of site $\mathbf{i}$ and $\mathbf{m_i}$ is the magnetic-vector field at that point. Two features are easily noted. First, the structure factors for different $t'$ approach the same value as $T \to 0$, which agrees with the Hartree-Fock approximation at low temperature. Then, the rise in $S(\mathbf{Q})$, which is indicative of the onset of long-range AFM order, becomes sharper with increasing $t'$ because $T_N$ gets smaller. It may be recalled that $S(\mathbf{Q})$ remains largely unaffected by the system size except in the vicinity of $T \sim T_N$ [54].

Fig. 1(b) shows the onset of short-range magnetic order defined by

$$\phi_1 = \frac{1}{4N} \sum_{\langle i, j \rangle} \langle \mathbf{m}_i \cdot \mathbf{m}_j \rangle, \tag{11}$$

where $\langle i, j \rangle$ denotes summation over nearest neighbors. It appears that $\phi_1$ is not independent of $t'$. In particular, it diminishes with a rise in $t'$ when $T \gtrsim T_N$. Below $T_N$, however, the behavior of short- and long-range magnetic order is similar. Therefore, an important question arises, is non vanishing of the short-range magnetic correlation function linked to the

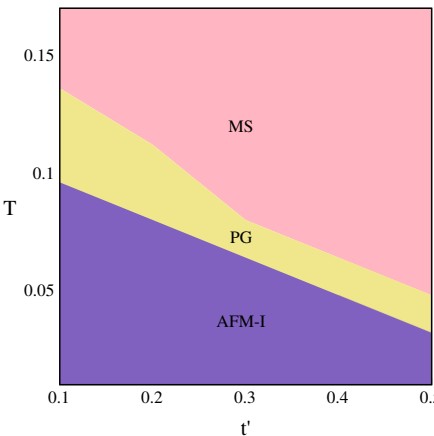

Figure 3: $t'-T$ phase diagram based on the onset temperatures $T_N$ and $T^*$ described in the main text, where paramagnetic metallic (PM), pseudogap-like (PG) and AFM-insulating (AFM-I) phases are shown. The region occupied by the pseudogap-like phase is reduced with increasing $t'$ indicating antagonistic behavior between the two.

pseudogap-like features in the spectral function? Perhaps, the nature of rise in the structure factor in the vicinity of $T_N$ as well as short-range magnetic correlations can be an indicator for the pseudogap-like feature. As we will see below that a sharper rise in the structure factor indicates a smaller temperature window for the pseudogap-like features and vice-versa.

Fig. 2 shows the density of states (DOS) calculated for $T/T_N = 1$ and 1.5 using

$$N(\omega) = \sum_{\mathbf{q},\lambda,\mathbf{i}} |\psi_{\mathbf{q},\lambda}(\mathbf{i})|^2 \delta\left(\omega - E_{\mathbf{q},\lambda}\right). \tag{12}$$

Here, $E_{\mathbf{q},\lambda}$ and $\psi_{\mathbf{q},\lambda}$ are the eigenvalues and eigenvectors for the whole superlattice. It is not difficult to notice the persisting pseudogap-like dip in the DOS at the Néel temperature and beyond. However, most interestingly, the dip becomes shallower with rising $t'$ and it can be seen to be almost absent for $t' \sim 0.4$ near $T/T_N \sim 1.5$ and beyond, whereas it is present at $T/T_N \sim 1$. Thus, the next-nearest neighbor hopping appears to be unfavorable for the pseudogap-like features beyond $T_N$. This can be noticed also in the $t'-T$ phase diagram (Fig. 3) where the temperature windows for both the long-range AFM order as well as for the pseudogap-like phase shrinks with rising $t'$. We have chosen onset temperature of the AFM order to be the temperature where $S(\mathbf{Q})$ starts to rise from zero. Similarly, the onset temperature $T^*$ of the pseudogap-like phase, marked by presence of dip in the density of state, is determined by the condition when there is no further change in the dip of the DOS or the dip disappears as temperature rises.

Next, we examine the evolution of quasiparticle dispersion as a function of temperature using the single-particle spectral function

$$A(\mathbf{k},\omega) = \sum_{\mathbf{q},\lambda} |\langle \mathbf{k} | \psi_{\mathbf{q},\lambda} \rangle|^2 \delta\left(\omega - E_{\mathbf{q},\lambda}\right), \tag{13}$$

where $\langle \mathbf{k} | \psi_{\mathbf{q},\alpha} \rangle = \sum_l \sum_i \langle \mathbf{k} | l, i \rangle \langle l, i | \psi_{\mathbf{q},\lambda} \rangle$, and $l, i$ are superlattice and site indices, respectively. Fig. 4 shows the quasiparticle dispersion for $t'/t = 0.3$ a value close to the one obtained through various estimates for high-$T_c$ cuprates [30, 60, 61]. Well-formed but asymmetrical gap can be seen near $(\pi/2, \pi/2)$ as well as $(\pi, 0)$. The hole- and particle-excitation energies are the least near $(\pi/2, \pi/2)$ and $(\pi, 0)$, respectively. The gap does not disappear even

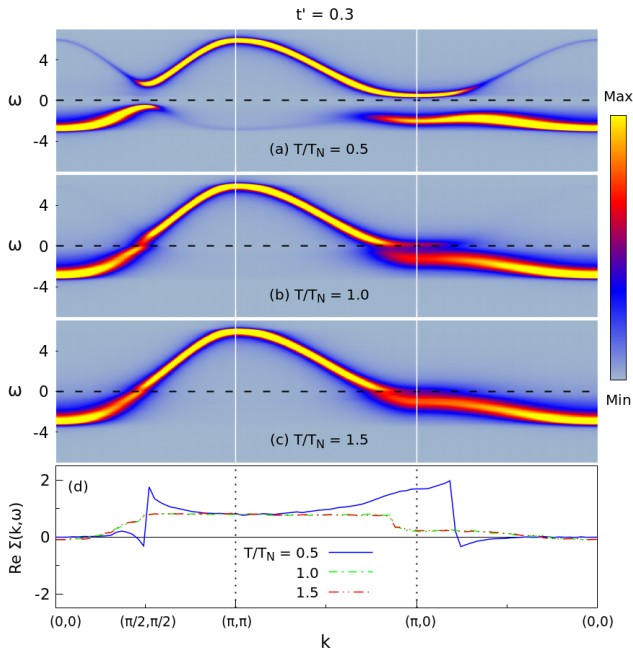

Figure 4: Quasiparticle dispersion for $t' = 0.3$ along the high symmetry directions for three different temperatures at $T/T_N =$ (a) 0.5, (b) 1.0, and (c) 1.5. Dashed line in (a-c) is corresponds to the Fermi level. (d) Real part of self-energy at different temperatures.

at $T/T_N = 1$ and beyond, which is evident from the suppression of spectral weight at the Fermi level. More specifically, the gap disappears near $(\pi/2, \pi/2)$ relatively more quickly with rising temperature and it can be seen to persist near $(\pi, 0)$ even at a relatively higher temperature. Another band with a relatively smaller spectral weight is found along $(0,0) \to (\pi, 0)$, $(0,0) \to (\pi/2, \pi/2)$, and $(\pi/2, \pi/2) \to (\pi, 0)$, which disappears near $T_N$ and beyond. The real part of the self-energy correction, below as well as beyond $T_N$, is significant in the region $(\pi/2, \pi/2) \to (\pi, \pi) \to (\pi, 0)$ (Fig. 4(d)). In order to compare the quasiparticle spectral weight along the high-symmetry directions, we also plot $A(\mathbf{k}, 0)$ as shown in the Fig. 5. The spectral weight is the largest near $(\pi/2, \pi/2)$ and $(\pi, 0)$, with an increasing peak height with $t'$. Elsewhere, it is suppressed to be negligibly small. In addition, the peaks height changes drastically near $T/T_N \sim 1$ as expected but continues to have a very small but non zero value even below $T/T_N = 1$ in contrast with the Hartree-Fock mean-field theories. Fig. 6 shows $A(\mathbf{k}, 0)$ in the entire Brillouin zone for the non-interacting case as well as for interacting case at different temperatures. A significant broadening is found all along the normal state Fermi surface arising because of the thermal fluctuations in the order-parameter fields. In addition, the correlation induced modification of the Fermi surface, which is significant in the vicinity of $(\pi, 0)$, can also be noticed.

Further, we look at the evolution of the gap structure along the normal-state Fermi surfaces as a function of temperature for different $t'$ (Fig. 7). At lower $t' = 0.1$, the asymmetry in the particle-hole excitation is weak, however, the gap along the normal-state Fermi surface survives at $T_N$ and beyond. On the contrary, the particle-hole asymmetry is much more evident for a relatively larger $t' \sim 0.3$ as expected. Near $(\pi/2, \pi/2)$, the hole excitation energy is significantly smaller in comparison to the particle excitation energy. As one moves along $(\pi/2, \pi/2) \to (\pi, 0)$, the hole-excitation energy increases while the particle-excitation energy decreases. Secondly, there is a significant asymmetry in the quasiparticle peak size. It is largest

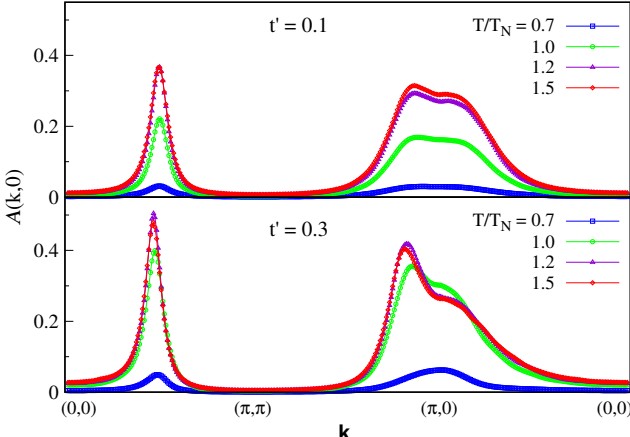

Figure 5: $A(\mathbf{k}, 0)$ along the high-symmetry directions for $t' = $ (a) 0.1 and (b) 0.3 at several different temperatures $T/T_N = 0.7$, 1.0, 1.2 and 1.5.

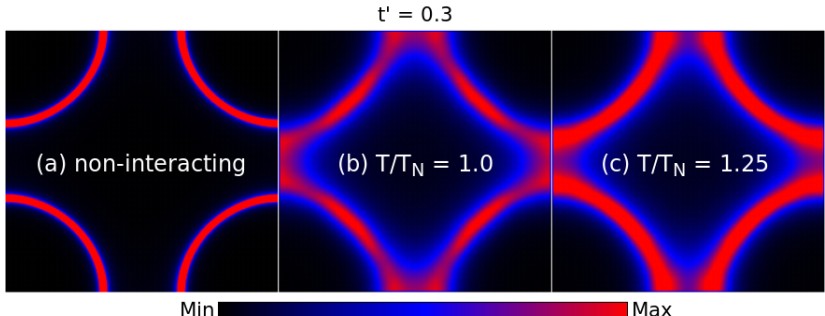

Figure 6: $A(\mathbf{k}, 0)$ for $t' = 0.3$ in the entire Brillouin zone for (a) non-interacting case, (b) $T = T_N$, and (c) $T = 1.25 T_N$.

for the hole excitation near $(\pi/2, \pi/2)$ in comparison to the particle excitation. These features are reversed as one moves towards $(\pi, 0)$.

Fig. 8 shows the hole-excitation energy along the normal state Fermi surface as a function of next-nearest neighbor hopping. The excitation energy increases monotonically on moving from $(\pi/2, \pi/2)$ to $(\pi, 0)$ for all $t'$ and becomes almost linear in the vicinity of $T/T_N \sim 1$. It is also nearly independent of $t'$ in the vicinity of $(\pi/2, \pi/2)$. The energy decreases with a rise in temperature as the spectral weight continues to get transferred to the Fermi level. For $T \gtrsim T_N$ and higher $t'$, the energy nearly vanishes.

# 4   Discussion

One important consequence of inclusion of $t'$ is the shift of spectral weight to higher and lower values of the quasiparticle energy depending on whether $t'$ induced energy change $4t' \cos k_x \cos k_y$ is positive or negative for a given quasiparticle momentum. This feature can be seen in our results, especially when $T < T_N$ and the lower and upper bands are accompanied with significant suppression of spectral weight in parts of the high-symmetry direction. The spectral weight is shifted towards the Fermi level near $(\pi, 0)$ and away to a large energy near $(\pi, \pi)$ or $(0, 0)$. These features are in agreement with the results obtained via cluster-perturbation theory [62]. More importantly, our calculation establishes that the gap at the

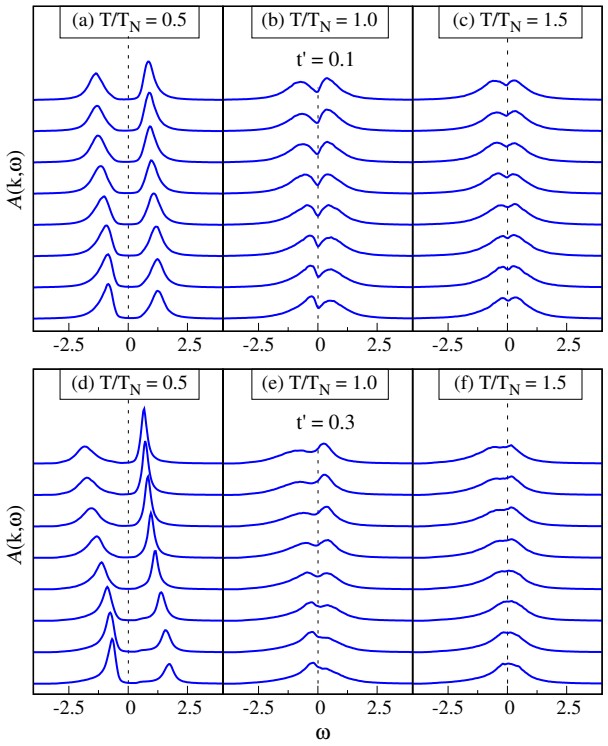

Figure 7: $A(\mathbf{k}, \omega)$ as a function of $\omega$ for (a-c) $t' = 0.1$ and 0.3 (d-f) at three different temperatures $T/T_N = 0.5, 1.0, 1.5$. The curves at the bottom and top correspond to the points $(\pi/2, \pi/2)$ and $(\pi, 0)$, respectively, while the others to the points in between as one moves from $(\pi/2, \pi/2)$ to $(\pi, 0)$ along the normal-state Fermi surface.

Fermi level does not disappear near $T \sim T_N$ and beyond though the gain in the spectral weight does take place with rising temperature.

In this work, we have restricted our study to the half-filled Hubbard model, which corresponds to zero doping. However, majority of the theoretical and experimental works have focused on the hole-doped cuprates because that leads to the appearance of unconventional high-$T_c$ $d$-wave superconductivity. The doping, however, introduces not only the $d$-wave superconductivity but a variety of other complex phases including the nematic, pair-density wave, striped spin and charge order, charge-density wave etc. The origin of these phases in the hole-doped cuprates are yet to be completely understood. On the other hand, the simulations that we applied to the half-filled Hubbard model provide us with important insight into the role of next-nearest neighbor hopping with regard to the spectral features.

At the half filling, we ignored the spatial and thermal fluctuations in the auxiliary field corresponding to the charge degree of freedom. This was based on the assumption that the effect of the fluctuations in the magnetic moments, is expected to be comparatively large on the single-particle spectral function. This is mainly because the rotation of the magnetic moments costs only a small amount of energy in comparison to the double occupancy involved with the charge fluctuations at half filling. In other words, the charge fluctuations are suppressed unless the system is doped with the charge carriers. For a finite hole doping, the effective model used in the simulation should suitably modified in order to study the magnetic and charge dynamics away from half filling. One such related model is the $t$-$J$ model [63] with explicit two terms describing magnetic-exchange and charge-density interaction. As indicated earlier, the temporal fluctuations are ignored in our approach, which results into the absence of the Brinkman-Rice peak of the quasiparticle excitation near the Mott transition for a moderate

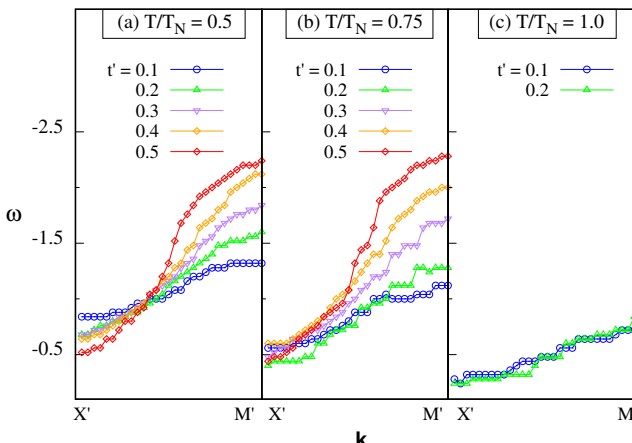

Figure 8: The hole-excitation energy calculated with the help of spectral function $A(\mathbf{k}, \omega)$ as one moves on the Fermi surface from along $X' = (\pi/2, \pi/2) \rightarrow M' = (\pi, 0)$.

$U$, otherwise obtained by the DMFT-based approaches. This is at the cost of full inclusion of the spatial fluctuations in the auxiliary fields and their consequences on the single-particle spectral function.

Findings on the role of next-nearest neighbor hopping $t'$ shows a very good qualitative agreement of momentum-dependent spectral features with experiments for the undoped cuprates [12, 14, 16]. Interestingly, these features are qualitatively similar to what are observed for the hole-doped cuprates especially in the direction $(\pi/2, \pi/2) \rightarrow (\pi, 0)$ along the normal state Fermi surface in the pseudogap and $d$-wave superconducting phase. Within the scheme used in the current work, study of spectral function for the hole-doped case will necessarily involve at least the competition between two tendencies, i.e., AFM ordering and $d$-wave superconducting, which, in turn, will require the inclusion of auxiliary fields associated with $d$-wave superconductivity also.

The consequence of competing interactions, in the case of doped cuprates, can be examined within either $t$-$J$ model [63] or $t$-$U$-$V$ models [64]. It will be of strong interest to see the consequence of such a competition on the momentum-dependent gap structure in the $d$-wave state as it will help to find the answer to questions such as does the $d$-wave gap get enhanced because of the AFM ordering tendencies? Answer to that question may help in gaining insight into the role of $t'$ in increasing the superconducting-transition temperature. Here, it may be recalled that our findings also indicate momentum-dependent gap structure at higher temperature even in the absence of any long-range magnetic order. We also find that the temperature window, where the pseudogap-like features exists, shrinks with an increase in $t'$. This raises another pertinent question about the compatibility of the pseudogap phase with $t'$ while it may be noted that a larger $t'$ is known to enhance $T_c$.

## 5 Conclusion

To conclude, we have examined in details, role of the next-nearest neighbor hopping on the single-particle excitation near AFM ordering temperature and beyond. Our findings based on an approach free of any finite size effect, while taking into account the thermal and spatial fluctuations, provides important insight into nature of possible hole and particle excitations along the high-symmetry direction in the half-filled Hubbard model. The spectral gap along high-symmetry, which persists even beyond AFM ordering temperature, shows a very good

qualitative agreement with the experiments on undoped cuprates whereas the results also indicate that the long-range hopping may not be favorable for the pseudogap phase. On the other hand, the increase in the gap size along the high symmetry direction especially along nodal to anti-nodal point often used in the context of cuprate supercondctors, grows with an increase in the next-nearest neighbor hopping. Here, we have restricted ourself to the case of only one auxiliary field corresponding to the magnetic moment. However, the approach adopted in this work can suitably be modified to incorporate other auxiliary fields. These auxiliary fields may correspond to the $d$-wave superconductivity or to the charge-density wave. Such a study of momentum-resolved spectral function incorporating multiple auxiliary fields can provide critical insight into the origin of the pseudogap phase observed in the high-$T_c$ cuprates.

# Acknowledgments

**Funding information** D.K.S. was supported through DST/NSM/R&D HPC Applications/2021/14 and PARAM Brahma supercomputing facility funded by DST-NSM and start-up research grant SRG/2020/002144 funded by DST-SERB.

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
