# Peer review of "Effect of next-nearest neighbor hopping on the single-particle excitations at finite temperature"

_SciPost Physics, doi:SciPost Phys. 16, 107 (2024)_

## Round 1 · Referee Report · Anonymous (Referee 1) · 2023-11-16

Report

In this theoretical work the authors analyze the single-orbital Hubbard model at half-filling using a theoretical scheme exploiting exact-diagonalization and Monte-Carlo. Their focus is to investigate the role of nearest-neighbor (nn) hopping on the single particle spectral function at different temperature. The analysis reveals a strong asymmetry of the hole/particle-exitation energy along the high-symmetry direction. The coherence features present a strong dependence from the momentum and appear to be compatible with the pseudogap phase of cuprates. The range of temperature in which those are observed within the model shrinks with increasing the nn hopping, thus nn may not be supportive to these pseudogap-like features.
The results are discussed at half-filling and considering only the magnetic fluctuating channel. The conclusion of the work well discuss what to expect by doping the system and point out the importance of taking into account multiple interacting channel in order to explore the role of the nn hopping in a more realistic model.

The topic of the research is of high interest for the community working on correlated systems and superconductivity. It provides interesting insights and hints that could be used in the future to further explore the role of nn hopping e.g. in doped systems, in the presence of other interacting channels and so on. The quality of the research is very high and the presentation of the results is very good. The technical aspects of the computation are explicitly discussed highlighting the advantage of the choose procedure and the approximations involved in the calculation.

I find the paper fulfilling the criteria for publication, however I invite the authors to consider a few comments before resubmitting.

Requested changes

1) The introduction is currently organized via a back and forth from theory and experiments to discuss the two main aspects that this work wants to address, i.e. the asymmetry of the phase diagram and the temperature and momentum-dependence of the quasiparticle excitations. I think it would be helpful to separate the two so that the reader will have first a clear idea of the experimental aspects that will be considered followed by an overview of the theoretical state of the art in this context. I invite the authors to consider such a reorganization of the introduction.

2) The description of Fig. 7 is misleading. In the text the authors claim that the raising of energy moving from (pi/2, pi/2) to (pi, 0) is almost linear, however the plots show a more complex behavior (those curve are not even close straight lines). The only temperature at which the energy curves are lines is at T~ TN were the grows is also quite small. I think that what describe the plot saying that the increasing is monotonic would generate less confusion.

3) Again in Fig 7. Can the author find a x-label easier to understand looking at the plot withour reading the caption? One option would be putting explicitly (pi/2, pi/2) and (pi, 0) as stating and ending point.

  • validity: high
  • significance: good
  • originality: high
  • clarity: high
  • formatting: excellent
  • grammar: excellent

Author:  Dheeraj Kumar Singh  on 2024-01-18  [id 4260]

(in reply to Report 1 on 2023-11-16)
Category:
answer to question

We are thankful to the referee for the encouraging comments on the current work. We also appreciate specific comments, that have been helpful in improving the manuscript further.

Comment: The introduction is currently organized via a back and forth from theory and experiments to discuss the two main aspects that this work wants to address, i.e. the asymmetry of the phase diagram and the temperature and momentum-dependence of the quasiparticle excitations. I think it would be helpful to separate the two so that the reader will have first a clear idea of the experimental aspects that will be considered followed by an overview of the theoretical state of the art in this context. I invite the authors to consider such a reorganization of the introduction.

Reply: We have now restructured the introduction so that the parts giving the theoretical and experimental backgrounds are now separate, begining with the experimental aspects first, and then followed by an overview of the theoretical status.

Comment: The description of Fig. 7 is misleading. In the text the authors claim that the raising of energy moving from $(\pi/2,\pi/2)$ to $(\pi,0)$ is almost linear, however the plots show a more complex behavior (those curve are not even close straight lines). The only temperature at which the energy curves are lines is at $T~ T_N$ were the grows is also quite small. I think that what describe the plot saying that the increasing is monotonic would generate less confusion.

Reply: We have modified the corresponding sentence in the manuscript so that there is no confusion. Now, instead, it is mentioned that the hole-excitation energy increases monotonically upon moving from $(\pi/2,\pi/2)$ toward $(\pi,0)$ and it varies almost linearly in the vicinity of $T_N$.

Comment: Again in Fig 7. Can the author find a x-label easier to understand looking at the plot without reading the caption? One option would be putting explicitly $(\pi/2,\pi/2)$ and $(\pi,0)$ as stating and ending point.} \\

Reply: We have now used symbols, which are frequently employed to represent the high-symmetry points in the Brillouin zone. In particular, it may be noted that $(\pi/2,\pi/2)$ is also one of the high symmetry points in the reduced or magnetic Brillouin zone. Thus, $(\pi/2,\pi/2)$ and $(\pi,0)$ are represented by X$^{\prime}$ and M$^{\prime}$, respectively. We hope that with this change, the $x$-label will now be easier to understand.

---

## Round 1 · Referee Report · Anonymous (Referee 2) · 2023-12-12

Report

In the present manuscript the authors provide an analysis of the effects induced by a nearest-neighbor hopping amplitude on the half filled Hubbard model. In particular, they focus on the spectral properties along the node and antinode in the pseudogap regime. The present studies extends a previous work of the same authors (Ref. 34) to a finite t'. They conclude that larga values of t' are unfavorable to the pseudogap behavior.
The method they used combines several approaches, and involves a series of limitations. Specifically, the method can not be extended to finite doping because the charge degrees of freedom are frozen (both spatial and thermal fluctuations are neglected). Similarly, the treatment of d-wave superconducting fluctuations is not included. In the conclusions, the authors state that additional auxiliary fields can be introduced, details are however missing. Given the restricted applicability of the approach, I would have appreciated a paragraph or short section about a concrete route that allows to overcome the above restrictions. I also think the limitations of the method should be mentioned in the methods section and not only in the discussion at the end.
The findings on the t' dependence are interesting and worth to be published, but given the limitations of the approach I would rather recommend SciPost Physics Core as suitable journal, after the points raised have been addressed.
  • validity: -
  • significance: -
  • originality: -
  • clarity: -
  • formatting: -
  • grammar: -

Author:  Dheeraj Kumar Singh  on 2024-01-18  [id 4261]

(in reply to Report 2 on 2023-12-12)
Category:
answer to question

General comment: In the present manuscript the authors provide an analysis of the effects induced by a nearest-neighbor hopping amplitude on the half filled Hubbard model. In particular, they focus on the spectral properties along the node and antinode in the pseudogap regime. The present studies extends a previous work of the same authors (Ref. 34) to a finite $t^\prime$. They conclude that large values of $t^\prime$ are unfavorable to the pseudogap behavior.

Reply: We are thankful to the referee for various constructive comments. In the following, we reply to specific comments by listing them pointwise.

Comment: The method they used combines several approaches, and involves a series of limitations. Specifically, the method can not be extended to finite doping because the charge degrees of freedom are frozen (both spatial and thermal fluctuations are neglected).

Reply: (i) We would like to clarify that the method can be extended to finite doping, however, the effective model should be modified accordingly. An appropriate effective model will be $t-J$ model, which is a derived from the Hubbard model. The $d$-wave superconductivity can arise from the next-nearest neighbor attractive interactions present in the $t-J$ model and magnetic order will arise from the nearest neighbor exchange interaction. Alternatively, one can use $t-U-V$ model as well to investigate the consequence of competing $d$wave superconductivity, where $U$ and $V$ are on-site repulsive and next-nearest neighbor attractive interaction parameters.

(ii) In the current work, we were primarily interested in investigating the role of next-nearest neighbor hopping on the single-particle spectrum at half filling, because, to the best of our knowledge, this problem has remained largely unexplored due to the challenges arising from (a) small cluster size limiting the momentum resolution and (b) famous sign problem in the implementation of simulations based on quantum Monte Carlo (QMC) or determinant QMC. Notably, these methods go beyond the mean-field approximation, and can incorporate the thermal and spatial fluctuations.

(iii) Furthermore, at half filling, the effect of the thermal fluctuations in the order-parameter field, which corresponds to the magnetic moments in the current work, is expected to be significantly large on the single-particle spectral function in comparison to the charge fluctuations. This is because a small rotation of magnetic moment does not require as much energy as involved in the charge fluctuations corresponding to the double occupancy of a site, unless the system is doped with the charge carriers.

Comment: Similarly, the treatment of $d$-wave superconducting fluctuations is not included. In the conclusions, the authors state that additional auxiliary fields can be introduced, details are however missing.

Reply: As we mentioned earlier that we were primarily focused on the role of next-nearest neighbor hopping on the single-particle spectrum. It would be of strong interest to know how the presence of two competing order such as magnetic and $d$-wave superconductivity affect the spectral features, particularly, the pseudogap-like features. In the discussion, we have a added a brief discussion as to how the additional auxiliary fields can be introduced via $t-J$ model, which is a derived from the Hubbard model for small dopings. The $d$-wave superconductivity can arise from the next-nearest neighbor attractive interactions present in the $t-J$ model. Alternatively, one can use $t-U-V$ model as well to investigate the consequence of competing $d$wave superconductivity, where $U$ and $V$ are on-site repulsive and next-nearest neighbor attractive interaction parameters.

Comment: Given the restricted applicability of the approach, I would have appreciated a paragraph or short section about a concrete route that allows to overcome the above restrictions. I also think the limitations of the method should be mentioned in the methods section and not only in the discussion at the end.

Reply: We have incorporated suggested discussion as well as mentioned the limitations of the method in the method section.

Comment: The findings on the $t^\prime$ dependence are interesting and worth to be published, but given the limitations of the approach I would rather recommend SciPost Physics Core as suitable journal, after the points raised have been addressed.

Reply: Our earlier work was mostly focused on the development and testing of the new method which combined three techniques namely, traveling-cluster approximation, parallelization of the update process and twisted-boundary conditions. The sole aim of combining these three techniques is the accessibility to a large-system size critical for a better momentum resolution for the spectral function so that any conclusion on the nature of spectral gap may be free from any finite size effect. Then, we demonstrated this idea with the help of half-filled Hubbard model with only nearest-neighbor hopping.

However, the actual correlated-electron system described by an effective one-band model may involve long-range hopping. The system of high-$T_c$ cuprates is one such standard example. Presence of the long-range hopping can have significant impact not only on the phase diagram but also on the spectral properties etc. which deserves a separate treatment. Several studies in the past have attempted to investigate the role of $t^{\prime}$. However, the approaches mainly include Hartree-Fock approximation based mean-field theory, dynamical mean-field theory (DMFT) etc., which do not incorporate the spatial fluctuations in the order-parameter field. The former does not even take into account the thermal fluctuations in the order-parameter field.

On the other hand, the methods such as QMC, detQMC etc., which go beyond mean-field theories, also use finite cluster size and suffer from famous sign problem in the absence of the particle-hole symmetry, \textit{i. e.} when the long-range hopping is considered. Even if doping is introduced, the use of these methods may be restricted to a certain temperature range.

In our simulation, in the absence of particle-hole symmetry, the chemical potential may fluctuate with temperature, therefore, the simulation requires to check the chemical potential and update the same in accordance with half filling at intermediate steps. This enhances the computational cost in the simulation process. Another factor which is responsible for raising the computational cost is the frustration introduced by the next-nearest neighbor hopping in the system, which raises the equiliberation time.

To the best of our knowledge, the current work may be the first study, which has examined the temperature dependent role of next-nearest neighbor hopping in the half-filled Hubbard model on the spectral features with a momentum resolution which is almost free from finite-size effect. For all the above reasons, we believe that the manuscript may deserve the visibility associated with SciPost Physics.

Anonymous on 2024-01-18  [id 4263]

(in reply to Dheeraj Kumar Singh on 2024-01-18 [id 4261])

With these explanations, I now more clearly understand the points made by the authors and am convinced of the relevance of the present contribution. The manuscript appears clearly improved by the changes and the additional discussions. I recommend publication.

---

## Round 1 · Referee Report · Anonymous (Referee 3) · 2023-12-23

Report

In the present manuscript, the authors discuss the effect of next-nearest neighbor (NNN) hopping t' on the spectral function of the half-filled single-band Hubbard model on the square lattice at finite temperature, with a special focus devoted to 'pseuodogap-like' features.
Their method of choice is an approach that corresponds to a mean-field treatment of the interaction term, combining exact diagonalization for the electronic part with a Monte Carlo sampling of the magnetic vector fields of the magnetization. At low temperatures, the technique reduces to standard Hartree-Fock approximation. It notably ignores spatial and thermal fluctuations in the charge channel and also does not include any temporal fluctuations. These limitations restrict its applicability to the half-filled case of the model. The technique has been described in the authors' previous publication, Ref. 34, where they also provide a very similar analysis of the same quantities of interest for the case t'=0.

Overall, the paper is providing some new insights on the modifications of the spectral function of the Hubbard model with respect to t' via this mean-field method. However, the pseudogap phase of the doped and undoped Hubbard model has been studied in detail over the last decades. A detailed comparison to known results for the 'pseudogap' phase at the half-filled t-t' Hubbard model as well as on the relation to other mean-field techniques is missing. It is therefore not clear for the readers, where the paper contributes novel findings and where it is simply reproducing results that are already well-known, even though now accessed with a potentially different numerical technique. Furthermore, some statements in the paper are inprecise or misleading and definitely need to be corrected, see below.

For all these reasons, I am not sure whether the manuscript meets the acceptance criteria of the journal and cannot recommend the publication of the manuscript in its current form in SciPost Physics. Some of the most urgent points that would need revision before an eventual resubmission are listed below.

Requested changes

1)- A detailed comparison with the existing literature at half-filling is missing. Only stating that the findings are similar to those of cluster perturbation theory is not sufficient. It is unclear which features are in (qualitative or quantitative) agreement with CPT. Furthermore, there are many techniques which provided results that allow for comparison, some of them going beyond the limitations of the mean-field technique used here, some being at a comparable level. These differences need to be discussed. In particular, the authors might want to compare their results to numerically exact results on the PG phase of the Hubbard model, e.g. using lattice Quantum Monte Carlo (QMC) , determinant Quantum Monte Carlo (detQMC) or Variational quantum Monte Carlo (VMC).
2)- Also the technique itself should be discussed in view of other techniques, in particular of mean-field type. How does it differ from standard mean-field techniques, slave bosons, slave spins, Hubbard-1 approximation etc.
3)- It is unclear why the discussion focusses on the high-symmetry path $(\pi/2,\pi/2)-(\pi,0)$. Changing t' results in a change of the dispersion on the non-interacting level, $U=0$, and reshapes the Fermi surface. It is thereby clear that some changes of the spectral function are simply due to the modifications of the dispersion, the shift of van-Hove singularities etc. In order to assess the effects of interactions, it would be useful to provide the non-interacting dispersion or the Fermi surface. To make statements on the 'pseudogap-like' features of the spectral function, its behavior should be studied along the Fermi surface with respect to the non-interacting reference for each t'.
4)- Referring to the dip in the density of states (DOS) as a 'pseudogap-like feature' can be misleading. The pseudogap is a k-dependent phenomenon that does not necessarily need to be related to a dip in the DOS. Such a dip can in principle also stem from a (fully) gapped system at low temperature, whose gap is progressively filled by thermal excitations at higher temperature. The authors' technique includes spatial fluctuations in the spin sector, thereby they should have access to the k-dependent self-energy which can give them a precise (and direct) answer on the existence or absence of a pseudogap.
5)- Figure 1: A legend should be added to make clear which quantities are plotted in panels a) and b) - the spin structure factor and the NN spin-spin correlator.
6)- Figures 4 & 5 seem to show the very same data, plotted in a color map and a line plot respectively. Figure 5 should therefore be removed or replaced by a plot which shows the amplitude of the spectral function along the high symmetry path for the different temperatures.
7)- The presentation of Figures 6 & 7 is slightly confusing and needs to be improved, in particular depicting the k-path more clearly. Also, it is not clear why the path $(\pi/2,\pi/2)-(\pi,0)$ can be compared in this way as a function of t', which even modifies the dispersion of the non-interacting model.
8)- The whole study is done at $U=4t$, which is less than half the bandwidth of the system. Nevertheless, the authors seem to suggest that they can safely interpret the broadening of the spectral function in therms of the Heisenberg spin exchange $J=4t^2/U$. This is problematic for different reasons: i) $U=4t$ is far from the Heisenberg limit; ii) the NNN spin exchange J' is not taken into account, despite the presence of the NNN hopping t'; iii) the broadening of the spectral function should be rather described by (the imaginary part of) the self-energy, which the authors do not show.
9)- Claims on agreement with experiment need to be substantiated. Which cuprates, which measurement techniques and which studies do the authors have in mind? What does it mean to be "in good agreement with experiment"?
10)- In the discussion and conclusion, the authors should be careful in assessing the transferability of their results to doped cuprates: 1) Their technique cannot be applied in a straight-forward way to doped systems since it neglects several types of fluctuations, which are known to be important in these systems. 2) The pseudogap of doped systems is not necessarily the same pseudogap that the authors study here at half-filling with a mean-field approach that is taylored to capture the physics in the strong-coupling limit.
11)- Given the vast literature on the t-t' Hubbard model at half-filling, the authors should explain more carefully what they mean when saying that their study fills a 'long-standing gap'.

  • validity: ok
  • significance: ok
  • originality: ok
  • clarity: ok
  • formatting: reasonable
  • grammar: reasonable

Author:  Dheeraj Kumar Singh  on 2024-01-18  [id 4262]

(in reply to Report 3 on 2023-12-23)

We are thankful to the referee for various constructive comments. Below, we respond to those comments pointwise while the corresponding changes in the revised manuscript is also mentioned.

Comment: A detailed comparison with the existing literature at half-filling is missing. Only stating that the findings are similar to those of cluster perturbation theory is not sufficient. It is unclear which features are in (qualitative or quantitative) agreement with CPT. Furthermore, there are many techniques which provided results that allow for comparison, some of them going beyond the limitations of the mean-field technique used here, some being at a comparable level. These differences need to be discussed. In particular, the authors might want to compare their results to numerically exact results on the PG phase of the Hubbard model, e.g. using lattice Quantum Monte Carlo (QMC), determinant Quantum Monte Carlo (detQMC) or Variational quantum Monte Carlo (VMC).

Reply: We would like to clarify that the exact-diagonalization + Monte-Carlo (ED+MC) based method used in the current work incorporates all types of thermal and spatial fluctuations in the order-parameter fields associated with the antiferromagnetic order. This is in contrast with the standard meanfield theories which ignore both thermal and spatial quantum fluctuations. Thus the magnetic moments, unlike the static mean-field theory, does not melt in the method used here when the system approaches $T_N$. Similarly, in the Hartree-Fock approximation based approaches, the spectral function can develop a gap only below $T_N$. On the contrary, we find that the gap persists beyond $T_N$, which we refer to pseudogap-like gap. Another advantage of the current approach is that as the temperature increases, thermal fluctuations start dominating over the quantum fluctuations, therefore, the accuracy of ED + MC increases in capturing essential physics.

A variety of techniques have been utilized which goes beyond standard meanfield theories, which mainly include density-matrix renormalization group (DMRG), dynamical-meanfield approximation (DMFT), cluster DMFT, QMC, detQMC, VMC, etc. The applicability of DMRG has largely been restricted to the quasi-one dimensional systems. Although, the DMFT-based methods do capture the Mott transition, the self-energy correlation incorporated is independent of momentum rendering it not suitable for the study of momentum-dependent spectral features. The QMC and detQMC were used extensively to study finite-size system with the difficulty that, at low temperatures, the correlations length is greater than the lattice size. Therefore, the correlations are overestimated for smaller clusters because they are artificially closer to criticality than a system in the thermodynamic limit. This may result in the disentanglement of the MI and AFM transitions. Another important consequence of the finite-size effect is a good momentum resolution for the single-particle spectral function, which has remained challenging till now.

Another important issue with QMC and DetQMC is the sign problem, which severly restricts its range of applicability especially when the hopping beyond nearest-neighbors is considered, which introduces particle-hole asymmetry. Even when doping away from half-filling is considered, the DetQMC can used only in a limited temperature range.
For these reasons, despite a significant volume of work at half filling, most of them were restricted to only nearest-neighbor hopping, particularly, the studies based QMC, detQMC, etc. The effect of $t^{\prime} \ne 0$ has been mainly studied using cluster-perturbation theory (CPT). CPT combines the solutions of small individual clusters of an infinite lattice system with Block theory of conventional band description to provide an approximation for the Green's function in the thermodynamic limit. In particular, at intermediate interaction strength, it may be challenging to make accurate prediction for the Hubbard gap mainly constrained by finite-size induced level splitting. We have incorporated the summary of all the points discussed above in the revised manuscript.

Comment:Also the technique itself should be discussed in view of other techniques, in particular of mean-field type. How does it differ from standard mean-field techniques, slave bosons, slave spins, Hubbard-1 approximation etc.

Reply: Slave-bosons or -spins mean-field theoretic approaches have been quite successful in studying the Mott transition. Recent developments also show their uses in studying the magnetically ordered phases. However, again inclusion of magnetic order ignores the spatial and thermal fluctuation, which is contrastingly different from the method we adopt in this work.

Comment: It is unclear why the discussion focuses on the high-symmetry path $(\pi/2,\pi/2)-(\pi,0)$. Changing $t^\prime$ results in a change of the dispersion on the non-interacting level, $U=0$, and reshapes the Fermi surface. It is thereby clear that some changes of the spectral function are simply due to the modifications of the dispersion, the shift of van-Hove singularities etc. In order to assess the effects of interactions, it would be useful to provide the non-interacting dispersion or the Fermi surface. To make statements on the 'pseudogap-like' features of the spectral function, its behavior should be studied along the Fermi surface with respect to the non-interacting reference for each $t^\prime$.

Reply: We were primarily focused on how the gap opens along the Fermi surface. While the Fig. 4 and 5 showed the dispersion with quasiparticle momentum-resolved amplitude, Fig. 6 and 7 showed how the gap evolves along the Fermi surface. In Fig. 7, there is typo, the high-symmetry direction $(\pi/2,\pi/2)-(\pi,0)$ should be replaced with Fermi surface along $(\pi/2,\pi/2)-(\pi,0)$. Along the Fermi surface, gapless excitation is present, therefore, we did not plot it. We have corrected the typo, that is, replaced the phrase ``high-symmetry direction''
by ``Fermi surface $(\pi/2,\pi/2) \rightarrow (\pi,0)$''.

Comment: Referring to the dip in the density of states (DOS) as a 'pseudogap-like feature' can be misleading. The pseudogap is a $k$-dependent phenomenon that does not necessarily need to be related to a dip in the DOS. Such a dip can in principle also stem from a (fully) gapped system at low temperature, whose gap is progressively filled by thermal excitations at higher temperature. The authors' technique includes spatial fluctuations in the spin sector, thereby they should have access to the $k$-dependent self-energy which can give them a precise (and direct) answer on the existence or absence of a pseudogap.

Reply: In using the phrase 'pseudogap-like feature' for the current context, we have followed the convention used for the hole-doped cuprates. In other words, with rising temperature, the gap should be filled up. Conventionally, it is expected that the gap should disappear as the temperatures increases up to $T_N$. As seen from Fig. 2, the gap continues to persist despite the loss of long-range order, which is also reflected in the momentum-resolved spectral function. In such a scenario, we needed a terminology to describe this phenomenon. For this reason, we have used the phrase.

Comment:Figure 1: A legend should be added to make clear which quantities are plotted in panels a) and b) - the spin structure factor and the NN spin-spin correlator.

Reply: In the revised manuscript, we have added the legends separately to the Fig. 1 (a) and (b) to avoid any confusion.

Comment: Figures 4 \& 5 seem to show the very same data, plotted in a color map and a line plot respectively. Figure 5 should therefore be removed or replaced by a plot which shows the amplitude of the spectral function along the high symmetry path for the different temperatures.

Reply: We have removed Fig. 5 and included a Figure showing the amplitude of the spectral function along the high symmetry path at different temperatures.

Comment:The presentation of Figures 6 \& 7 is slightly confusing and needs to be improved, in particular depicting the $k$-path more clearly. Also, it is not clear why the path $(\pi/2,\pi/2)-(\pi,0)$ can be compared in this way as a function of $t^\prime$, which even modifies the dispersion of the non-interacting model.

Reply: The confusion here arises because of a typo. Actually, the ${\bf k}$ path in both the figures are along the Fermi surface along the path $(\pi/2,\pi/2)-(\pi,0)$ instead along the high-symmetry direction along $(\pi/2,\pi/2)-(\pi,0)$. We have modified the caption to remove this confusion.

Comment: The whole study is done at $U=4t$, which is less than half the bandwidth of the system. Nevertheless, the authors seem to suggest that they can safely interpret the broadening of the spectral function in therms of the Heisenberg spin exchange $J = 4t^2\/U$. This is problematic for different reasons: i) $U=4t$ is far from the Heisenberg limit; ii) the NNN spin exchange $J^\prime$ is not taken into account, despite the presence of the NNN hopping $t^\prime$; iii) the broadening of the spectral function should be rather described by (the imaginary part of) the self-energy, which the authors do not show.

Reply: It is true that $U=4t$ is far from the strong coupling limit where the Hubbard model can be mapped to the Heisenberg model. Here, we would like to clarify that though the results are presented in the manuscript only for $U = 4t$, we did check the calculations for higher value of $U$ also and found that the thermal broadening increases with increasing $U$. For this reason, we wanted to comment as to what is expected when $U$ is increased. When incorporated NNN spin exchange $J^\prime$ because of NNN hopping $t^\prime$, the broadening will be enhanced further because of the frustration introduced.

Comment: Claims on agreement with experiment need to be substantiated. Which cuprates, which measurement techniques and which studies do the authors have in mind? What does it mean to be "in good agreement with experiment"?

Reply: In the current work, we are primarily interested in the effect of next-nearest neighbor hopping on the single-particle excitation of half-filled Hubbard model. Therefore, the phrase "in good agreement with experiment" was used in reference to the ARPES measurements carried out in the undoped cuprates, particularly, the momentum-dependent single-particle gap sructures (PRL 74, 964 (1995), PRL 80, 4245 (1998), PRB 70, 092503
(2004)).

Comment: In the discussion and conclusion, the authors should be careful in assessing the transferability of their results to doped cuprates: 1) Their technique cannot be applied in a straight-forward way to doped systems since it neglects several types of fluctuations, which are known to be important in these systems. 2) The pseudogap of doped systems is not necessarily the same pseudogap that the authors study here at half-filling with a mean-field approach that is tailored to capture the physics in the strong-coupling limit.

Reply: We agree that our results are not transferable to the doped cuprates because then several types of order-parameter fields will come into picture and the $d$-wave superconducting order parameter is prominent amongst them. It is indeed true that the pseudogap feature that we discuss may be entirely different from the one arising as a result of multiple competing orders in the doped cuprates. Precisely, for this reason, we have used the phrase ``pseudogap-like'' instead of psuedogap at various points in the manuscript.

Comment: Given the vast literature on the $t-t^\prime$ Hubbard model at half-filling, the authors should explain more carefully what they mean when saying that their study fills a 'long-standing gap'.

Reply: As discussed in the reply to earlier comments, to the best of our understanding, most of the earlier studies, which go beyond mean-field theory, have focused on the half-filled Hubbard model without next-nearest neighbor hopping mainly because of particle-hole asymmetry induced sign problem. Furthermore, CDMFT or CPT may provide a picture corresponding only to small clusters, thus suffering from finite-size induced level splitting. On the other hand, the method that we have used for the simulation, beside being free from sign problem, has the advantage of accessing a large-sized system, thus able to obtain a momentum-resolution never obtained before. A very good momentum resolution, which is free from finite-size effect, is absolutely necessary to conclusively establish the existence of small gap as found in the pseudogap or pseudogap-like phases along the Fermi surface. It is in this respect, we used the phrase 'long-standing gap'. We have modified the phrase in the revised manuscript to avoid any emphasis on the ``long-standing gap''.

---

## Round 2 · Author Response

Reply to the comments of Referees Comments of referee 1

We are thankful to the referee for the encouraging comments on the current work. We also appreciate specific comments, that have been helpful in improving the manuscript further.

  1. Comment: The introduction is currently organized via a back and forth from theory and experiments to discuss the two main aspects that this work wants to address, i.e. the asymmetry of the phase diagram and the temperature and momentum-dependence of the quasiparticle excitations. I think it would be helpful to separate the two so that the reader will have first a clear idea of the experimental aspects that will be considered followed by an overview of the theoretical state of the art in this context. I invite the authors to consider such a reorganization of the introduction.

  2. Reply: We have now restructured the introduction so that the parts giving the theoretical and experimental backgrounds are now separate, begining with the experimental aspects first, and then followed by an overview of the theoretical status.

  3. Comment: The description of Fig. 7 is misleading. In the text the authors claim that the raising of energy moving from $(\pi/2,\pi/2)$ to $(\pi,0)$ is almost linear, however the plots show a more complex behavior (those curve are not even close straight lines). The only temperature at which the energy curves are lines is at $T~ T_N$ were the grows is also quite small. I think that what describe the plot saying that the increasing is monotonic would generate less confusion.

Reply: We have modified the corresponding sentence in the manuscript so that there is no confusion. Now, instead, it is mentioned that the hole-excitation energy increases monotonically upon moving from $(\pi/2,\pi/2)$ toward $(\pi,0)$ and it varies almost linearly in the vicinity of $T_N$.

  1. Comment: Again in Fig 7. Can the author find a x-label easier to understand looking at the plot without reading the caption? One option would be putting explicitly $(\pi/2,\pi/2)$ and $(\pi,0)$ as stating and ending point.

Reply:We have now used symbols, which are frequently employed to represent the high-symmetry points in the Brillouin zone. In particular, it may be noted that $(\pi/2,\pi/2)$ is also one of the high symmetry points in the reduced or magnetic Brillouin zone. Thus, $(\pi/2,\pi/2)$ and $(\pi,0)$ are represented by X$^{\prime}$ and M$^{\prime}$, respectively. We hope that with this change, the $x$-label will now be easier to understand.

Comments of referee 2

General comment In the present manuscript the authors provide an analysis of the effects induced by a nearest-neighbor hopping amplitude on the half filled Hubbard model. In particular, they focus on the spectral properties along the node and antinode in the pseudogap regime. The present studies extends a previous work of the same authors (Ref. 34) to a finite $t^\prime$. They conclude that large values of $t^\prime$ are unfavorable to the pseudogap behavior.

Reply: We are thankful to the referee for various constructive comments. In the following, we reply to specific comments by listing them pointwise.

(i). Comment: The method they used combines several approaches, and involves a series of limitations. Specifically, the method can not be extended to finite doping because the charge degrees of freedom are frozen (both spatial and thermal fluctuations are neglected).

Reply: (1) We would like to clarify that the method can be extended to finite doping, however, the effective model should be modified accordingly. An appropriate effective model will be $t-J$ model, which is a derived from the Hubbard model. The $d$-wave superconductivity can arise from the next-nearest neighbor attractive interactions present in the $t-J$ model and magnetic order will arise from the nearest neighbor exchange interaction. Alternatively, one can use $t-U-V$ model as well to investigate the consequence of competing $d$wave superconductivity, where $U$ and $V$ are on-site repulsive and next-nearest neighbor attractive interaction parameters.

(2) In the current work, we were primarily interested in investigating the role of next-nearest neighbor hopping on the single-particle spectrum at half filling, because, to the best of our knowledge, this problem has remained largely unexplored due to the challenges arising from (a) small cluster size limiting the momentum resolution and (b) famous sign problem in the implementation of simulations based on quantum Monte Carlo (QMC) or determinant QMC. Notably, these methods go beyond the mean-field approximation, and can incorporate the thermal and spatial fluctuations.

(3) Furthermore, at half filling, the effect of the thermal fluctuations in the order-parameter field, which corresponds to the magnetic moments in the current work, is expected to be significantly large on the single-particle spectral function in comparison to the charge fluctuations. This is because a small rotation of magnetic moment does not require as much energy as involved in the charge fluctuations corresponding to the double occupancy of a site, unless the system is doped with the charge carriers.

(ii). Comment: Similarly, the treatment of $d$-wave superconducting fluctuations is not included. In the conclusions, the authors state that additional auxiliary fields can be introduced, details are however missing.

Reply: As we mentioned earlier that we were primarily focused on the role of next-nearest neighbor hopping on the single-particle spectrum. It would be of strong interest to know how the presence of two competing order such as magnetic and $d$-wave superconductivity affect the spectral features, particularly, the pseudogap-like features. In the discussion, we have a added a brief discussion as to how the additional auxiliary fields can be introduced via $t-J$ model, which is a derived from the Hubbard model for small dopings. The $d$-wave superconductivity can arise from the next-nearest neighbor attractive interactions present in the $t-J$ model. Alternatively, one can use $t-U-V$ model as well to investigate the consequence of competing $d$wave superconductivity, where $U$ and $V$ are on-site repulsive and next-nearest neighbor attractive interaction parameters.

(iii). Comment: Given the restricted applicability of the approach, I would have appreciated a paragraph or short section about a concrete route that allows to overcome the above restrictions. I also think the limitations of the method should be mentioned in the methods section and not only in the discussion at the end.

Reply: We have incorporated suggested discussion as well as mentioned the limitations of the method in the method section.\

(iv). Comment: The findings on the $t^\prime$ dependence are interesting and worth to be published, but given the limitations of the approach I would rather recommend SciPost Physics Core as suitable journal, after the points raised have been addressed.

Reply: Our earlier work was mostly focused on the development and testing of the new method which combined three techniques namely, traveling-cluster approximation, parallelization of the update process and twisted-boundary conditions. The sole aim of combining these three techniques is the accessibility to a large-system size critical for a better momentum resolution for the spectral function so that any conclusion on the nature of spectral gap may be free from any finite size effect. Then, we demonstrated this idea with the help of half-filled Hubbard model with only nearest-neighbor hopping.

However, the actual correlated-electron system described by an effective one-band model may involve long-range hopping. The system of high-$T_c$ cuprates is one such standard example. Presence of the long-range hopping can have significant impact not only on the phase diagram but also on the spectral properties etc. which deserves a separate treatment. Several studies in the past have attempted to investigate the role of $t^{\prime}$. However, the approaches mainly include Hartree-Fock approximation based mean-field theory, dynamical mean-field theory (DMFT) etc., which do not incorporate the spatial fluctuations in the order-parameter field. The former does not even take into account the thermal fluctuations in the order-parameter field.

On the other hand, the methods such as QMC, detQMC etc., which go beyond mean-field theories, also use finite cluster size and suffer from famous sign problem in the absence of the particle-hole symmetry, \textit{i. e.} when the long-range hopping is considered. Even if doping is introduced, the use of these methods may be restricted to a certain temperature range.

In our simulation, in the absence of particle-hole symmetry, the chemical potential may fluctuate with temperature, therefore, the simulation requires to check the chemical potential and update the same in accordance with half filling at intermediate steps. This enhances the computational cost in the simulation process. Another factor which is responsible for raising the computational cost is the frustration introduced by the next-nearest neighbor hopping in the system, which raises the equiliberation time.

To the best of our knowledge, the current work may be the first study, which has examined the temperature dependent role of next-nearest neighbor hopping in the half-filled Hubbard model on the spectral features with a momentum resolution which is almost free from finite-size effect. For all the above reasons, we believe that the manuscript may deserve the visibility associated with SciPost Physics.

Comments of referee 3 We are thankful to the referee for various constructive comments. Below, we respond to those comments pointwise while the corresponding changes in the revised manuscript is also mentioned.

  1. Comment: A detailed comparison with the existing literature at half-filling is missing. Only stating that the findings are similar to those of cluster perturbation theory is not sufficient. It is unclear which features are in (qualitative or quantitative) agreement with CPT. Furthermore, there are many techniques which provided results that allow for comparison, some of them going beyond the limitations of the mean-field technique used here, some being at a comparable level. These differences need to be discussed. In particular, the authors might want to compare their results to numerically exact results on the PG phase of the Hubbard model, e.g. using lattice Quantum Monte Carlo (QMC), determinant Quantum Monte Carlo (detQMC) or Variational quantum Monte Carlo (VMC).

Reply: We would like to clarify that the exact-diagonalization + Monte-Carlo (ED+MC) based method used in the current work incorporates all types of thermal and spatial fluctuations in the order-parameter fields associated with the antiferromagnetic order. This is in contrast with the standard meanfield theories which ignore both thermal and spatial quantum fluctuations. Thus the magnetic moments, unlike the static mean-field theory, does not melt in the method used here when the system approaches $T_N$. Similarly, in the Hartree-Fock approximation based approaches, the spectral function can develop a gap only below $T_N$. On the contrary, we find that the gap persists beyond $T_N$, which we refer to pseudogap-like gap. Another advantage of the current approach is that as the temperature increases, thermal fluctuations start dominating over the quantum fluctuations, therefore, the accuracy of ED + MC increases in capturing essential physics.

A variety of techniques have been utilized which goes beyond standard meanfield theories, which mainly include density-matrix renormalization group (DMRG), dynamical-meanfield approximation (DMFT), cluster DMFT, QMC or detQMC, VMC, etc. The applicability of DMRG has largely been restricted to the quasi-one dimensional systems. Although, the DMFT-based methods do capture the Mott transition, the self-energy correlation incorporated is independent of momentum rendering it not suitable for the study of momentum-dependent spectral features. The QMC or detQMC were used extensively to study finite-size system with the difficulty that, at low temperatures, the correlations length is greater than the lattice size. Therefore, the correlations are overestimated for smaller clusters because they are artificially closer to criticality than a system in the thermodynamic limit. This may result in the disentanglement of the MI and AFM transitions. Another important consequence of the finite-size effect is a good momentum resolution for the single-particle spectral function, which has remained challenging till now.

Another important issue with QMC or DetQMC is the sign problem, which severly restricts its range of applicability especially when the hopping beyond nearest-neighbors is considered, which introduces particle-hole asymmetry. Even when doping away from half-filling is considered, the DetQMC can used only in a limited temperature range. For these reasons, despite a significant volume of work at half filling, most of them were restricted to only nearest-neighbor hopping, particularly, the studies based QMC, detQMC, etc. The effect of $t^{\prime} \ne 0$ has been mainly studied using cluster-perturbation theory (CPT). CPT combines the solutions of small individual clusters of an infinite lattice system with Block theory of conventional band description to provide an approximation for the Green's function in the thermodynamic limit. In particular, at intermediate interaction strength, it may be challenging to make accurate prediction for the Hubbard gap mainly constrained by finite-size induced level splitting. We have incorporated the summary of all the points discussed above in the revised manuscript.

  1. Comment: Also the technique itself should be discussed in view of other techniques, in particular of mean-field type. How does it differ from standard mean-field techniques, slave bosons, slave spins, Hubbard-1 approximation etc.

Reply: Slave-bosons or -spins mean-field theoretic approaches have been quite successful in studying the Mott transition. Recent developments also show their uses in studying the magnetically ordered phases. However, again inclusion of magnetic order ignores the spatial and thermal fluctuation, which is contrastingly different from the method we adopt in this work.

  1. Comment: It is unclear why the discussion focuses on the high-symmetry path $(\pi/2,\pi/2)-(\pi,0)$. Changing $t^\prime$ results in a change of the dispersion on the non-interacting level, $U=0$, and reshapes the Fermi surface. It is thereby clear that some changes of the spectral function are simply due to the modifications of the dispersion, the shift of van-Hove singularities etc. In order to assess the effects of interactions, it would be useful to provide the non-interacting dispersion or the Fermi surface. To make statements on the 'pseudogap-like' features of the spectral function, its behavior should be studied along the Fermi surface with respect to the non-interacting reference for each $t^\prime$.

Reply:We were primarily focused on how the gap opens along the Fermi surface. While the Fig. 4 and 5 showed the dispersion with quasiparticle momentum-resolved amplitude, Fig. 6 and 7 showed how the gap evolves along the Fermi surface. In Fig. 7, there is typo, the high-symmetry direction $(\pi/2,\pi/2)-(\pi,0)$ should be replaced with Fermi surface along $(\pi/2,\pi/2)-(\pi,0)$. Along the Fermi surface, gapless excitation is present, therefore, we did not plot it. We have corrected the typo, that is, replaced the phrase high-symmetry direction'' byFermi surface $(\pi/2,\pi/2) \rightarrow (\pi,0)$''.

  1. Comment: Referring to the dip in the density of states (DOS) as a 'pseudogap-like feature' can be misleading. The pseudogap is a $k$-dependent phenomenon that does not necessarily need to be related to a dip in the DOS. Such a dip can in principle also stem from a (fully) gapped system at low temperature, whose gap is progressively filled by thermal excitations at higher temperature. The authors' technique includes spatial fluctuations in the spin sector, thereby they should have access to the $k$-dependent self-energy which can give them a precise (and direct) answer on the existence or absence of a pseudogap.

Reply: In using the phrase 'pseudogap-like feature' for the current context, we have followed the convention used for the hole-doped cuprates. In other words, with rising temperature, the gap should be filled up. Conventionally, it is expected that the gap should disappear as the temperatures increases up to $T_N$. As seen from Fig. 2, the gap continues to persist despite the loss of long-range order, which is also reflected in the momentum-resolved spectral function. In such a scenario, we needed a terminology to describe this phenomenon. For this reason, we have used the phrase.

  1. Comment:Figure 1: A legend should be added to make clear which quantities are plotted in panels a) and b) - the spin structure factor and the NN spin-spin correlator.

Reply: In the revised manuscript, we have added the legends separately to the Fig. 1 (a) and (b) to avoid any confusion.

  1. Comment: Figures 4 \& 5 seem to show the very same data, plotted in a color map and a line plot respectively. Figure 5 should therefore be removed or replaced by a plot which shows the amplitude of the spectral function along the high symmetry path for the different temperatures.

Reply: We have removed Fig. 5 and included a Figure showing the amplitude of the spectral function along the high symmetry path at different temperatures.

  1. Comment: The presentation of Figures 6 \& 7 is slightly confusing and needs to be improved, in particular depicting the $k$-path more clearly. Also, it is not clear why the path $(\pi/2,\pi/2)-(\pi,0)$ can be compared in this way as a function of $t^\prime$, which even modifies the dispersion of the non-interacting model.

Reply: The confusion here arises because of a typo. Actually, the ${\bf k}$ path in both the figures are along the Fermi surface along the path $(\pi/2,\pi/2)-(\pi,0)$ instead along the high-symmetry direction along $(\pi/2,\pi/2)-(\pi,0)$. We have modified the caption to remove this confusion.

  1. Comment:The whole study is done at $U=4t$, which is less than half the bandwidth of the system. Nevertheless, the authors seem to suggest that they can safely interpret the broadening of the spectral function in therms of the Heisenberg spin exchange $J = 4t^2\/U$. This is problematic for different reasons: i) $U=4t$ is far from the Heisenberg limit; ii) the NNN spin exchange $J^\prime$ is not taken into account, despite the presence of the NNN hopping $t^\prime$; iii) the broadening of the spectral function should be rather described by (the imaginary part of) the self-energy, which the authors do not show.

Reply: It is true that $U=4t$ is far from the strong coupling limit where the Hubbard model can be mapped to the Heisenberg model. Here, we would like to clarify that though the results are presented in the manuscript only for $U = 4t$, we did check the calculations for higher value of $U$ also and found that the thermal broadening increases with increasing $U$. For this reason, we wanted to comment as to what is expected when $U$ is increased. When incorporated NNN spin exchange $J^\prime$ because of NNN hopping $t^\prime$, the broadening will be enhanced further because of the frustration introduced.

  1. Comment: Claims on agreement with experiment need to be substantiated. Which cuprates, which measurement techniques and which studies do the authors have in mind? What does it mean to be "in good agreement with experiment"?

Reply: In the current work, we are primarily interested in the effect of next-nearest neighbor hopping on the single-particle excitation of half-filled Hubbard model. Therefore, the phrase "in good agreement with experiment" was used in reference to the ARPES measurements carried out in the undoped cuprates, particularly, the momentum-dependent single-particle gap sructures (PRL 74, 964 (1995), PRL 80, 4245 (1998), PRB 70, 092503 (2004)).

  1. Comment:In the discussion and conclusion, the authors should be careful in assessing the transferability of their results to doped cuprates: 1) Their technique cannot be applied in a straight-forward way to doped systems since it neglects several types of fluctuations, which are known to be important in these systems. 2) The pseudogap of doped systems is not necessarily the same pseudogap that the authors study here at half-filling with a mean-field approach that is tailored to capture the physics in the strong-coupling limit.

Reply: We agree that our results are not transferable to the doped cuprates because then several types of order-parameter fields will come into picture and the $d$-wave superconducting order parameter is prominent amongst them. It is indeed true that the pseudogap feature that we discuss may be entirely different from the one arising as a result of multiple competing orders in the doped cuprates. Precisely, for this reason, we have used the phrase ``pseudogap-like'' instead of psuedogap at various points in the manuscript.

  1. Comment:Given the vast literature on the $t-t^\prime$ Hubbard model at half-filling, the authors should explain more carefully what they mean when saying that their study fills a 'long-standing gap'.

Reply: As discussed in the reply to earlier comments, to the best of our understanding, most of the earlier studies, which go beyond mean-field theory, have focused on the half-filled Hubbard model without next-nearest neighbor hopping mainly because of particle-hole asymmetry induced sign problem. Furthermore, CDMFT or CPT may provide a picture corresponding only to small clusters, thus suffering from finite-size induced level splitting. On the other hand, the method that we have used for the simulation, beside being free from sign problem, has the advantage of accessing a large-sized system, thus able to obtain a momentum-resolution never obtained before. A very good momentum resolution, which is free from finite-size effect, is absolutely necessary to conclusively establish the existence of small gap as found in the pseudogap or pseudogap-like phases along the Fermi surface. It is in this respect, we used the phrase 'long-standing gap'. We have modified the phrase in the revised manuscript to avoid any emphasis on the ``long-standing gap''.

---

## Round 2 · List of Changes

Highlights of major changes in the revised manuscript are given below while all the changes in the revised manuscript are indicated in blue color text.

(i) We have restructured the introduction in such a way that the experimental background is now followed by the theoretical background. Additional discussion is incorporated in reference to various comments.
(ii) Caption and discussion on Fig. 7 is modified.
(iii) A discussion on the limitation of the current approach as well as how to introduce additional auxiliary field is added.
(iv) As pointed out earlier, study of the role of next-nearest neigbhor hopping while going beyond mean-field theory, taking into account spatial and thermal fluctuations, and also being free from finite-size effect has not been carried out till date to the best knowledge of authors. Most of the earlier studies focused on Hubbard model with only nearest-neighbor hopping mainly because of sign problem in the simulation when particle-hole asymmetry is introduced via next-nearest neighbor hopping. The second issue with these method is finite size effect, which may put a limit to a good-momentum resolution in the spectral function. We have provided a related discussion in the introduction.
(v) The legend of Fig. 1 is modified.
(vi) Fig. 5 is replaced with another figure showing amplitude of the spectral function as a function of temperature.
(vii) Corresponding to new discussions in response to different comments, several new references are also incorporated.

---

## Round 3 · Referee Report · Anonymous (Referee 1) · 2024-1-30

Report

After considering the revised version of the manuscript and the reply from the authors to the first round of report I can recommend the article for publication.

As I already discussed in my previous report, the manuscript provides interesting insights and hints that could be used in the future to further explore the role of nn hopping e.g. in doped systems, in the presence of other interacting channels and so on. The quality of the research is very high and the results of the analysis are discussed in a convincing way. Technical aspects of the computation are explicitly discussed highlighting the advantage of the procedure used and the approximations involved in the calculation.

The current version of the manuscript presents a more logic introduction and better explain some aspects of the results that were unclear in the previous version of the manuscript.

---

## Round 3 · Referee Report · Anonymous (Referee 3) · 2024-2-1

Report

In their responses, the authors addressed all the points raised by the reviewers, almost all of them satisfactorily. In the revised version, most of the points have been taken into account and lead to changes that have improved the manuscript.
Nevertheless, I would have liked to have seen a representation of the self-energy that would have made it possible to compare the nature of the approximations of the authors' technique with other approaches based on Green's functions. Similarly, I still feel that plotting the Fermi surfaces of the non-interacting and interacting systems for different values of t' would have improved readability and made reading Figures 6 and 7 easier.
Since the manuscript has been much improved, including the correction of some misleading passages, adding a discussion of the limitations of the technique and providing a more complete reference to existing techniques, I recommend the article for publication in SciPost Physics.

---

## Round 3 · List of Changes

Followings are the list of changes made with respect to the previous version:
(i) Fig. 1 has been updated
(ii) The style of the manuscript has been changed to SciPost. The DOI has also been provided.

---

## Round 4 · Author Response

Reply to the comments of Referees
Comments of referee 1 (anonymous report 1)

Comment: After considering the revised version of the manuscript and the reply from the authors to the first round of report I can recommend the article for publication.
As I already discussed in my previous report, the manuscript provides interesting insights and hints that could be used in the future to further explore the role of nn hopping e.g. in doped systems, in the presence of other interacting channels and so on. The quality of the research is very high and the results of the analysis are discussed in a convincing way. Technical aspects of the computation are explicitly discussed highlighting the advantage of the procedure used and the approximations involved in the calculation.
The current version of the manuscript presents a more logic introduction and better explain some aspects of the results that were unclear in the previous version of the manuscript

Reply: We are thankful to the referee for several comments in the previous round of refereeing, which led to us to clarify many points by adding to the discussion part. This has improved the manuscript significangly and enhanced its readability.

Comments of referee 2 (anonymous report 2)

Comment: In their responses, the authors addressed all the points raised by the reviewers, almost all of them satisfactorily. In the revised version, most of the points have been taken into account and lead to changes that have improved the manuscript.
Nevertheless, I would have liked to have seen a representation of the self-energy that would have made it possible to compare the nature of the approximations of the authors' technique with other approaches based on Green's functions. Similarly, I still feel that plotting the Fermi surfaces of the non-interacting and interacting systems for different values of t' would have improved readability and made reading Figures 6 and 7 easier. Since the manuscript has been much improved, including the correction of some misleading passages, adding a discussion of the limitations of the technique and providing a more complete reference to existing techniques, I recommend the article for publication in SciPost Physics.

Reply: We express our thankfulness to the referee for several critical comments in the previous round of refereeing for the improvement of the manuscript. Accordingy, the revised version was significantly improved.

In the version, we are about to submit, we have incorporated the self energy into the Fig. 4 while the Fermi surface for interacting and non-interacting system is presented as Fig. 6. The real part of the self energy $\Sigma ({\bf k}, \omega)$ is plotted along the high-symmetry direction. The Fermi surface for the paramagnetic state with pseudogap-like feature has been plotted with the help of $A({\bf k}, 0)$ for temperature $T \ge T_N$ as it does not exist below $T_N$. A significant broadening of $A({\bf k}, 0)$ can be seen because of large thermal/spatial fluctuations in the order parameter fields at higher temperatures. It may be noted the Fermi surface is obtained with the help of $A({\bf k}, 0)$ with quasiparticle excitation energy being zero. On the other hand, a more general quantity such as $A({\bf k}, \omega)$ discussed in Fig. 6 and Fig. 7 of the previous version of the manuscript provides more detailed spectral features, especially the gap structures, of the single-particle excitation. Therefore, the consequences of variation of $t^{\prime}$ is mostly captured in Fig. 6 and Fig. 7 (Fig. 7 and Fig. 8 of the version to be submitted). A change of $t^{\prime}$ on the thermally broadened $A({\bf k}, 0)$ is expected to modify the Fermi surface as in the non-interacting case while retaining the interaction intruduced modification as in the case of $t^{\prime} = 0.3$. For this reason, we have restricted ourselves to the case of $t^{\prime} = 0.3$ and shown the temperature dependence $A({\bf k}, 0)$ instead.

---

## Round 4 · List of Changes

In accordance with the comments of second referee, we have provided the plot of the self energy as well as of Fermi surface for the non-interacting and interactin systems. One of the plots has added to the existing Fig. 4 and while another is introduced as Fig. 6, respectively. A brief discussion is also added in the related section and paragraph.

---

## Editorial Decision

published